# Effectiveness of Pre-Hospital Tourniquet in Emergency Patients with Major Trauma and Uncontrolled Haemorrhage: A Systematic Review and Meta-Analysis

**DOI:** 10.3390/ijerph182312861

**Published:** 2021-12-06

**Authors:** Roberto Latina, Laura Iacorossi, Alice Josephine Fauci, Annalisa Biffi, Greta Castellini, Daniela Coclite, Daniela D’Angelo, Silvia Gianola, Veronica Mari, Antonello Napoletano, Gloria Porcu, Matteo Ruggeri, Primiano Iannone, Osvaldo Chiara

**Affiliations:** 1National Centre for Clinical Excellence Healthcare Quality and Safety, Istituto Superiore di Sanità, Via Giano della Bella 34, 00162 Rome, Italy; roberto.latina@unipa.it (R.L.); laura.iacorossi@gmail.com (L.I.); alice.fauci@iss.it (A.J.F.); daniela.coclite@iss.it (D.C.); daniela.dangelo@iss.it (D.D.); veronica.mari@iss.it (V.M.); primiano.iannone@iss.it (P.I.); 2Department of Health Promotion Science, Maternal and Infant Care, Internal Medicine and Medical Specialities, University of Palermo, Piazza delle Cliniche 2, 90127 Palermo, Italy; 3IRCCS Regina Elena National Cancer Institute, Via Elio Chianesi 53, 00144 Roma, Italy; 4Laboratory of Healthcare Research and Pharmacoepidemiology, Department of Statistics and Quantitative Methods, University of Milano-Bicocca, Piazza dell’Ateneo Nuovo 1, 20126 Milan, Italy; annalisa.biffi@unimib.it (A.B.); gloria.porcu@unimib.it (G.P.); 5Unit of Clinical Epidemiology, IRCCS Istituto Ortopedico Galeazzi, Via Riccardo Galeazzi 4, 20161 Milan, Italy; greta.castellini@grupposandonato.it (G.C.); silvia.gianola@grupposandonato.it (S.G.); 6National Centre for Health Technology Assessment, Istituto Superiore di Sanità, Via Giano della Bella 34, 00162 Rome, Italy; matteo.ruggeri@iss.it; 7General Surgery and Trauma Team, ASST Grande Ospedale Metropolitano Niguarda, University of Milan, Piazza Ospedale Maggiore 3, 20162 Milan, Italy; osvaldo.chiara@unimi.it

**Keywords:** systematic review, tourniquet, haemorrhage, emergencies, meta-analysis

## Abstract

Trauma is one of the leading causes of uncontrolled haemorrhage, death, and disability. Use of a tourniquet can be considered an optimal anti-haemorrhagic resource, in pre-hospital and emergency settings, and its lifesaving effect is clinically contradictory. This review aims to assess the clinical efficacy of the tourniquet in the emergency pre-hospital care setting for the management of haemorrhage. We conducted the systematic review following the Preferred Reporting Items for Systematic Reviews and Meta-Analyses, the PRISMA statement. We searched the following electronic databases: EMBASE, MEDLINE, and Cochrane-CENTRAL. All studies included were appraised for risk of bias. Prevalent primary outcomes were mortality and use of blood products. Secondary outcomes were related to adverse effects. The quality of evidence was assessed using the Grading of Recommendations Assessment, Development and Evaluation approach (GRADE). Four studies were involved (1762 trauma patients). The adjusted odds ratio (aOR) of 0.47 (95% confidence Interval (CI) 0.19–1.16; three studies; 377 patients) for overall mortality estimates did not give a clear indication of the benefits of emergency pre-hospital tourniquets (PH-TQ) versus no pre-hospital tourniquet (NO PH-TQ) placement. The adjusted mean difference for blood product use was −3.28 (95% CI −11.22, 4.66) for packed red blood cells (pRBC) and −4.80 (95% CI −5.61, −3.99) for plasma, respectively. The certainty of evidence was downgraded to very low for all outcomes. Our results suggest an unclear effect of emergency pre-hospital tourniquet placement on overall mortality and blood product use. However, this systematic review highlights the availability of only observational studies and the absence of high quality RCTs assessing the efficacy of PH-TQs. Randomized controlled trials are needed.

## 1. Introduction

Every year, the lives of about 1.35 million people are cut short as a result of trauma caused by road traffic crashes, and between 20 and 50 million people worldwide suffer non-fatal injuries [1], with many incurring a disability [2]. Uncontrolled bleeding is the leading cause of death in 34% of trauma patients. Haemorrhage is one of the major causes of potentially preventable deaths in both civilian and military contexts [3]. The Italian National Institute for Statistics recorded a total of 172,183 traumas in 2019 [4], which, according to the Multi-Regional Serious Trauma Intra-Hospital Centre [5], was prevalent in the male population (75.7%), and 65.4% (*n* = 4385) required intensive care unit (ICU) hospitalization with a mean stay of 7.3 days (standard deviation = ±14.6), a fatal outcome of 27.5%, and 17.4% of patients requiring transfusions in the emergency room [6]. Bleeding control in major trauma is a clinical priority that can be achieved through either direct compression or the use of mechanical or pneumatic tourniquets (TQ). Direct pressure is the first and simplest step. TQs can be applied if direct pressure on the bleeding area is not sufficient to control the bleeding [7] and when direct pressure is not a feasible option. Direct pressure should be maintained by health care personnel, which is sometimes insufficient in number. Use of a TQ can be considered an optimal anti-haemorrhagic resource in an emergency setting, and its lifesaving effect has become more apparent [8]. The use and effectiveness of mechanical or pneumatic TQs appears to be associated with effective control of bleeding and lower mortality rates from bleeding [9]. This is why the Hartford Consensus Conference [10] encourages their widespread use among the civilian population for the early control of haemorrhage in the extremities (upper and lower limbs) when direct manual compression is ineffective or unfeasible. In the literature, the efficacy of TQ application for the management of bleeding in the pre-hospital phase in patients with major trauma is supported by relatively low, often contradictory evidence, or is even not recommended for use in the civilian context [8]. In particular, it remains an open question whether when used in the pre-hospital setting, it leads to better survival and a reduced need for blood, or blood component, transfusions. This systematic review aims to summarize current knowledge on the clinical efficacy of pre-hospital TQ, which has been used as a framework to support the development of Italian guidelines for pre-hospital emergency treatment of major trauma (MT) in civilian settings.

## 2. Materials and Methods

We conducted a systematic review to support the major trauma integrated management guideline panel of the Italian National Institute of Health (NIH) in formulating recommendations [11]. Following the GRADE-ADOPOLMENT methodology [12], and in accordance with the standards defined by the NIH [13], the multidisciplinary panel decided to apply a structured and systematic updating and adaptation process of their recommendations from the National Institute for Clinical Excellence (NICE) guideline NG39 on pre-hospital application of TQs [14]. We conducted the systematic review following the Preferred Reporting Items for Systematic Reviews and Meta-Analyses, the PRISMA statement [15]. Study protocol has been stored at the following link: https://osf.io/n526s/ (accessed date 29 November 2021). The research question for this systematic review was: “Is the use of pneumatic or mechanical tourniquets both clinically effective and cost-effective in improving outcomes in major trauma patients with haemorrhage?”.

### 2.1. Inclusion Criteria

We included randomized controlled trials (RCT); cohort studies with adjusted results for key confounders (e.g., injury severity, age, depth of shock, degree of head injury) or matched at baseline for these if no RCT was available and/or observational studies. Eligible studies should meet the following criteria: (1) population: children, young people and adults who have experienced a traumatic limb injury; (2) intervention: pneumatic and mechanical TQ use; (3) comparison: no TQ use; and (4) setting: pre-hospital. Studies including patients requiring massive transfusion resulting from civilian settings were included.

### 2.2. Outcome Measures and Follow-Up Assessment

The primary outcome measures selected for the analyses were: (i) 24 h-mortality, 30-day to 12-months mortality; (ii) volume of infused blood components; and (iii) health-related quality of life (e.g., Glasgow Coma Scale [GCS] score at discharge). The secondary outcomes referring to adverse effects were related: (iv) amputation; (v) nervous system disorder-paralysis; (vi) renal failure; and (vii) haemorrhage.

### 2.3. Search Strategy

We searched the following electronic databases: MEDLINE (Pub-Med), EMBASE, Cochrane Central Register of Controlled Trials (CENTRAL) with language restriction (English, Italian, Spanish, French, German), using and updating (from March 2015 up to February 2020) the search strategy of the high quality clinical guideline of NICE on MT [14], reported in Appendix A. We checked the reference lists of all studies included and of all the systematic reviews identified during the search process. We also searched for ongoing trials: ClinicalTrials.gov.

### 2.4. Study Selection and Data Extraction

Two independent authors (SG, GC) screened titles and abstracts according to the search strategy. Following the first phase, they independently assessed the full text of all potentially relevant studies for inclusion in this review. Any disagreement was resolved through discussion with a third author (OC). Then, using a standardized data collection form, the following information was extracted from the included studies: (i) study characteristics: study design, setting, countries and settings, funding; (ii) participants’ characteristics, sample size and type of trauma; and (iii) intervention type and outcomes. We contacted authors if the reported data were not sufficiently detailed or incomplete. Any discrepancy was resolved by consensus, or with the help of a third independent author (AB). The bibliographies of retrieved papers were also evaluated to identify other relevant articles to be included. Reasons for exclusion are reported in Figure 1.

### 2.5. Internal Validity

The internal validity of non-randomized studies was assessed using the Newcastle–Ottawa Scale [16]. The following domains were appraised: selection, comparability, and outcome. Thresholds for converting the Newcastle–Ottawa Scale to AHRQ standards (good, fair, and poor) were adapted. Two reviewers (SG, GC) independently evaluated the methodological quality of the included studies, and any disagreement was resolved by reaching a consensus between reviewers.

### 2.6. Data Synthesis

Whenever possible, meta-analyses were conducted to combine the outcome data using the DerSimonian and Kacker random effects model [17], which takes into account both the sampling variance within the studies and the variation in the underlying effect across studies due to the different populations and study designs. To assess the statistical heterogeneity across studies, we applied Cochran’s Q statistics and calculated the I^2^ test [18], using the following interpretation of the value of I^2^: 0 to 50 = low; 50 to 80 = moderate and worthy of investigation; 80 to 100 = severe and worthy of understanding; 95 to 100 = aggregate with major caution (Julian Higgins, personal communication). The analyses were performed using RevMan Version 5.4. The treatment effects for continuous outcomes were summarized as mean difference (MD) or standardized mean difference (SMD) when different outcome measurements were reported; the treatment effects for dichotomized outcomes were evaluated using the odds ratio (OR). When adjustments or propensity scores for each of the outcomes were available, we pooled them as adjusted odds ratios (aORs).

### 2.7. Quality of Evidence

The quality of evidence of each outcome was judged by evaluating five dimensions (risk of bias, consistency of effect, imprecision, indirectness, and publication bias) using the Grading of Recommendations Assessment Development and Evaluation (GRADE) approach [19]. The evidence was downgraded from ‘high quality’ by one level if serious limitations were found for each of the five dimensions, or by two levels, if very serious limitations were found. We presented a summary of findings describing the treatment effects, the quality of evidence, and the reasons for the limitations.

## 3. Results

A total of 395 records were screened. From the updated search, four observational studies [20,21,22,23] were included (Figure 1). For details on excluded studies, see Appendix A.

### 3.1. General Characteristics

None of the included studies were RCTs or a systematic review of RCTs. The four included studies were conducted in the U.S. and were retrospective cohort studies reporting data from trauma registers [20,21,22,23] and from computerized clinical data records [23]. The included studies allowed for the following comparisons: (1) pre-hospital tourniquet (PH-TQ) versus (vs.) tourniquet (NPH-TQ) [20,21,23]; and (2) pre-hospital tourniquet (TQ) vs. trauma centre-tourniquet (TC-TQ) [22]. The studies reviewed involved a total of 1762 trauma patients, 1455 of whom to compare PH-TQ application vs. NPH-TQ [20,21,23], and 306 to evaluate PH-TQ vs. TC-TQ application [22]. General characteristics of the studies are reported in Table 1. Demographic and clinical characteristics of observational studies are described in Table 2, Table 3 and Table 4.

### 3.2. Primary Outcomes

#### 3.2.1. Mortality at 24 h, 30 Days, and 12 Months

Three studies [20,21,23] compared PH-TQ intervention vs. NPH-TQ intervention and reported adjusted results on overall mortality (Table 3). Quantitative analyses were therefore performed on these only (Figure 2). One study [22], reported adjusted results on mortality due to trauma haemorrhage comparing PH-TQ vs. TC-TQ showing an odds ratio of 0.22 [CI 95% 0.06–0.80], and standard error = 0.66) [22] (Figure 3). The authors in [20,21,23] compared PH-TQ vs. non-intervention, or with intervention at the trauma centre, in terms of mortality caused by haemorrhage [22]. One study [20] was reported without an estimate of the effect, as the published data were not clearly reported.

Despite the differences between the observational studies included, the forest plots (Figure 2 and Figure 3) fairly consistently showed a lower mortality risk with pre-hospital application of the PH-TQ compared to NPH-TQ or TC-TQ. The analysis showed no differences between the experimental group that received TQ and the controls.

#### 3.2.2. Packed of Infused Blood Components

Three different types of blood components were analysed: (a) pRBC transfusion [20,21,22,23]; (b) Platelets [20,22] (Table 4); and (c) Plasma [20,22,23]. As shown in Table 4, Smith et al. (2019) provides an adjusted estimate showing how the application of the tourniquet reduces the number of transfused plasma units, whereas Texeira et al. (2018) did not provide clear indications about the benefits of applying the tourniquet on pRBC and plasma requirements (Figure 3).

#### 3.2.3. Length of Stay (LOS) in ICU

LOS in ICU in two study groups, PH-TQ group compared to the NPH-TQ and PH-TQ group compared to TC-TQ is reported as days in ICU [3] and 30-day ICU free days [21]. In the study by [22], the ICU LOS (mean days) was 3.9 (standard deviation (SD) = ±7.2 days) in the PH-TQ group vs. 2.9 (SD = ±6.5 days) (*p* = 0.064), OR (95% CI) = −1.01 (−2.09 to 0.06) in the control group. In the study by [22], the ICU length of stay (median days), using PH-TQ group compared to the trauma centre tourniquet was 0 (interquartile range (IQR) = 0.2) vs. 2 (IQR = 0.5), *p* < 0.01. In both studies, the time point is not defined. The study by [22] showed no clear difference in ICU LOS (days) between the PH-TQ group compared to NPH-TQ, 25.8 (SD = ±0.7 days) vs. 26.7 (SD = ±0.6) (*p*-value not significant), with a time point of 30 days (mean difference = −090[CI 95% = −2.71–0.91]).

#### 3.2.4. Health-Related Quality of Life

No study has reported on the outcomes of interest.

### 3.3. Secondary Outcomes

Four different secondary outcomes in terms of adverse events (Figure 4) reported from the following studies were classified:

Amputation: (i) Not reported whether early/late application: the study by [22] reports unadjusted values; (ii) initial application: the study by [21] shows how the application of the tourniquet increases the risk of initial amputation; and (iii) late application: three articles reported data on delayed placement [20,21,23], which provide an unclear estimate on the adjustments and are therefore not meta-analysable.

Nervous system disorders-paralysis: Only one article reported the data on nerve palsy [23], showing no significant differences between PH-TQ vs. NPH-TQ application.

Renal failure: One article reported data on renal failure showing no significant differences between PH-TQ vs. NPH-TQ application [21].

Haemorrhage: One article reported data on this adverse event showing [23], lower risk of performing procedures for the control of bleeding (due to a lower probability of bleeding) with the application of a tourniquet.

### 3.4. Internal Validity

Three studies were judged to be of good quality and one of fair quality (Appendix A). Overall, studies were affected by bias in selection and outcome domains.

### 3.5. Quality of Evidence

For all of the aforementioned outcomes, the certainty of evidence was downgraded to very low due to serious risk of bias, indirectness, and imprecision of the estimates (Appendix A). For the remaining outcomes (time to definitive control of haemorrhage and patient-reported outcomes), the quality of evidence was not assessed due to the absence of data.

## 4. Discussion

This systematic review highlights the efficacy of the emergency application of PH-TQ in civilian trauma. TQ application is on the rise in the USA, and this trend seems to be mainly in large cities or urban areas with well-developed trauma systems. To our knowledge, this study is the first to propose a meta-analysis of the results of the four retrospective cohort studies included.

### 4.1. Use of Tourniquet and Mortality

The studies reviewed about 1800 trauma patients, one third of whom received PH-TQ application. Our meta-analysis showed that use of PH-TQ did not appear to significantly influence mortality (aOR = 0.47, 95% CI 0.19–1.16; *p* = 0.55). The evidence (GRADE) on overall causes of mortality and mortality caused by haemorrhage was ranked as very low-certainty (critical). Our results are concordant with Scerbo et al. (2016) [24] (6% vs. 12%, *p* = 0.61). One recent review of ten studies failed to show a statistically significant difference in all-cause mortality where a tourniquet had been used compared with direct pressure alone [6]. Additionally, there was no difference in mortality outcomes in another study where tourniquets were applied in the presence or absence of shock [25]. Unsurprisingly, the early placement of a TQ in military casualty, before the onset of shock, was associated with a decreased risk of mortality [26]. Vascular injuries of the limbs in the military setting are common and this is probably the reason for the significant effect on survival. It is unclear whether the mortality rate in our population of reference is partially attributable to tourniquet use.

### 4.2. Blood Products Use and LOS in ICU

The aOR by meta-analysis for blood product use shows that PH-TQ use is significant for plasma (*p* < 0.001) and not for pRBC. Regarding the use of RBC in the first 24 h [23], it showed a reduction in the use of this type of blood component. Regarding plasma only, [22] demonstrated that blood transfusions within the first hour and mortality from haemorrhagic shock decreased among patients with PH-TQ use. However, compared to PH-TQ, patients with TC-TQ had a higher rate of transfusion within the first hour of arrival (*p* = 0.02). Moreover, Teixeira et al. [20] also explained that the tourniquet patients were more likely to require massive plasma transfusion, to present a higher Injury Severity Score (ISS) and Abbreviated Injury Scale (AIS) score, and to be in shock.

Regarding LOS in ICU, [22] provided an unclear estimate of the adjustment and therefore cannot be represented, and in [21], at 30-day ICU free days, no clear difference was perceptible between the two groups. Several data regarding TQ application should be interpreted carefully. A precise indication and definition of TQ exposure time have been rarely specified, as described in [6]. This may impact on the primary and secondary outcomes.

### 4.3. Secondary Outcomes

Moreover, unadjusted values on amputation can provide misleading information, and are therefore not meta-analysable; they do not provide a clear indication regarding the benefits of applying a tourniquet. Concerning haemorrhage, only a single article [23] indicates a lower risk of performing procedures to control bleeding due to a lower probability of bleeding when a tourniquet is applied, which describes aOR for adverse events of PH-TQ vs. NPH-TQ.

### 4.4. The Quality of Evidence

Regarding internal validity, although three studies were judged to be of good quality and one of fair quality (Appendix A), overall, studies were affected by bias in selection and outcome domains. Using GRADE methodology, the quality of evidence obtained from the observational studies for all of the aforementioned outcomes was downgraded to very low due to a serious risk of bias, indirectness, and imprecision of the estimates (Appendix A).

### 4.5. Implication for Clinical Practice

TQs were used in various situations and involved quite homogeneous types of injuries and locations; they were mainly used for extremity injuries. As described in [6], a precise indication and definition of TQ exposure time was rarely specified. Only one study defined the indications for TQ use as cases of extremity vascular injury [22]. Moreover, tourniquets should only be used to control extremity haemorrhage if direct pressure is not adequate or possible, for example, in the case of multiple victims or injuries, inaccessible wounds, or when nurses and medical staff are working together to achieve the resuscitation and stabilization of critical patients. Although not significantly, a trend toward reduction in mortality with PH-TQ was shown in our study. It is common sense that proximal TQ is able to stop ongoing haemorrhage of a limb and that the application in the field is a very simple action that reduces blood loss and improves haemodynamics. Moreover, after TQ application, the health care personnel are available for other manoeuvres while limb bleeding is controlled. TQ application should only be used if direct pressure is not adequate or possible. The available literature suggests that commercial tourniquets are superior to the application of direct manual pressure or haemostatic dressings for life-threatening limb bleeding [27]. We found that patients who received a PH-TQ were mostly men (range age 30–36 years), and that tourniquets were mainly applied by health staff, as described in a recent review [6]. The pre-hospital advanced emergency staff such as nurses and physicians (Italy does not have paramedics) should be familiar with a tourniquet that has been proven to be effective. There is evidence that staff trained to use PH-TQ are more likely to do so and have fewer fears of complications in comparison with untrained staff [28]. Requiring pre-hospital trauma life support certification at regular intervals should help with skill decline and continued education as this course is updated with the current evidence. Although in the majority of cases external haemorrhage will be controlled by employing a stepwise approach, TQ use could be considered as a potential anti-haemorrhagic resource, and its life-saving effect may become more apparent [8].

The results of this meta-analysis allowed us to determine a recommendation on the use of the tourniquet for trauma in a pre-hospital setting. The guideline becomes a country’s strongest tool for homogenizing the clinical behaviour of healthcare practitioners and monitoring adherence to recommendations.

## 5. Limits and Strength

This systematic review highlights the absence of high quality RCTs assessing the efficacy of PH-TQs. Quantitative analyses were performed on only three retrospective observational studies reporting adjusted results. Moreover, most of the included studies were affected by bias in selection and outcome domains, and the certainty of the evidence for all the assessed outcomes was judged as ‘very low’ due to a serious risk of bias, indirectness, and imprecision of the estimates. Finally, we have our concerns whether the results of the four included studies can all be combined into one conclusion, since all aforementioned arguments result in heterogeneous study populations. Despite these limitations, this study presented points of strength: the internal validity of the included studies, which was assessed using the Newcastle–Ottawa Scale for observational studies, was robust and largely acknowledged.

## 6. Conclusions

Based on observational data, our analysis suggests that the effectiveness of the early application of PH-TQ on the mortality rate in our population of reference, and on different blood components used, is unclear. Though this systematic review was unable to identify high quality evidence, the available evidence could be used by experts for formulating judgments and recommendations though a structured and transparent process such as the GRADE-ADOLOPMENT [12]. Adoption of a multi-centre registry with standardised prospective data collection, specific to tourniquet use, can serve to improve the trauma community’s understanding of the safety and effectiveness of tourniquet use in civilian trauma settings. Future studies, preferably randomised controlled trials, should be carried out in order to confirm preliminary results determined in observational studies.

## Figures and Tables

**Figure 1 ijerph-18-12861-f001:**
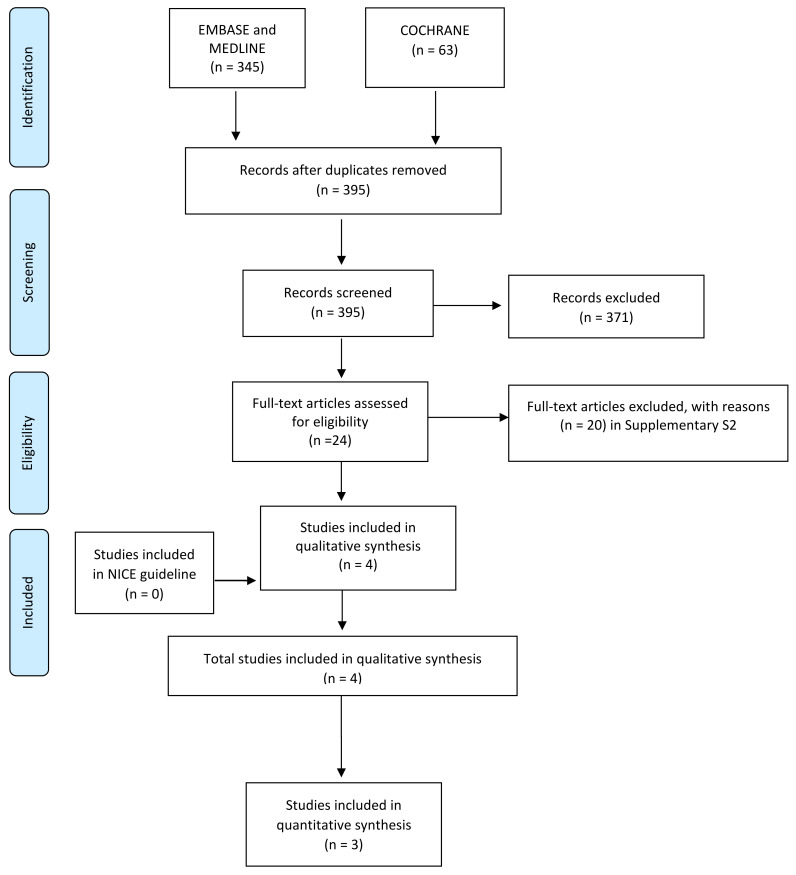
Flow chart of study selection.

**Figure 2 ijerph-18-12861-f002:**
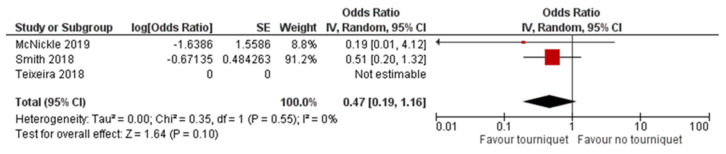
Adjusted odds ratio for overall mortality of pre-hospital-TQ vs. no pre-hospital-TQ.

**Figure 3 ijerph-18-12861-f003:**
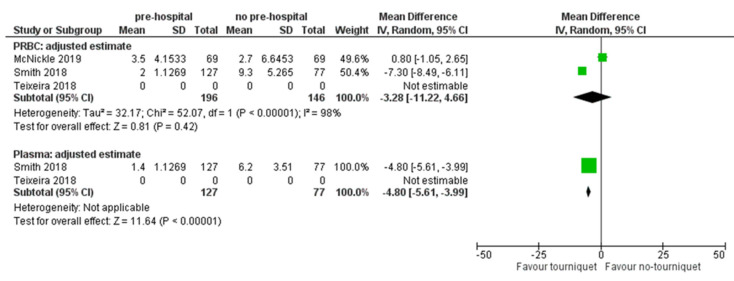
Adjusted odds ratios for mortality due to haemorrhage of PH-T vs. NO PH-T.

**Figure 4 ijerph-18-12861-f004:**
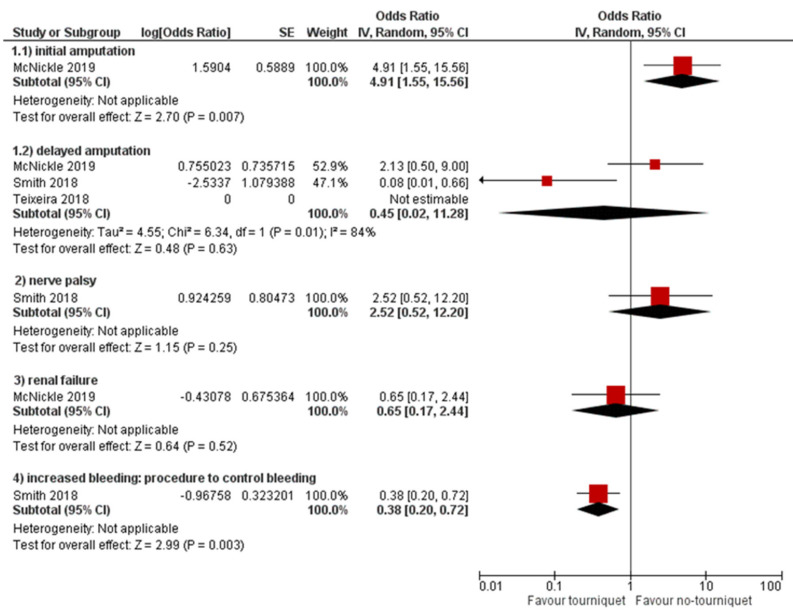
Adjusted odds ratio for adverse events of pre-hospital-TQ versus no pre-hospital-TQ.

**Table 1 ijerph-18-12861-t001:** General characteristics.

Study (First Author, Year)	Country (State)	Study Design	Setting	Patients (Total Sample)	Intervention	Comparison	Primary Outcomes	Secondary Outcomes
McNickle et al. (2019)		Retrospective cohort study	Level I trauma centre (Pre-hospital data)	192	Pre-hospital tourniquet application	No pre-hospital tourniquet application	Blood transfusions within the first 24 h	Hospital-free days Intensive care unit (ICU)-free Ventilator-free days (30-day benchmark) Presence of significant complications (acute kidney injury, rhabdomyolysis, compartment syndrome, limb loss) Mortality
Smith et al. (2019)	New Orleans	Retrospective cohort study	Level I trauma centre (Pre-hospital data)	238	Pre-hospital commercial tourniquet application for extremity injuries	No pre-hospital tourniquet application	Blood product utilization	Presence of shock on arrival Limb complications related to tourniquet use Systemic complications Hospital length of stay (LOS) ICU length of stay In-hospital mortality
Teixeira et al. (2018)	Texas	Multicentre retrospective cohort study	11 level I trauma centres (Pre-hospital data)	1026	Pre-hospital tourniquet application	No pre-hospital tourniquet application	In-hospital mortality	Delayed amputation Thrombo-embolic complications Respiratory complications Cardiac complications Infectious complications Hospital length of stay (LOS) ICU length of stay Ventilator days
Scerbo et al. (2017)	Texas	Retrospective cohort study	Memorial Hermann Hospital	306	Pre-hospital tourniquet application	Trauma centre tourniquet application	Death from haemorrhagic shock	Physiology on presentation to the TC Massive transfusion of blood products
							The outcomes were determined by a combination of Trauma Registry data and electronic health record review.	

**Table 2 ijerph-18-12861-t002:** Demographic and clinical characteristics of observational studies.

Study	Sample		Age (years)		Sex (Male)		ISS		Extremity AIS		GCS		HR		SBP	
	N		Mean (SE)		N(%)		Mean (SE)		Mean (SE)		Median (IRQ Range)		Mean (SE)		Mean (SE)	
	PH-T	NPH-T	PH-T	NPH-T	PH-T	NPH-T	PH-T	NPH-T	PH-T	NPH-T	PH-T	NPH-T	PH-T	NPH-T	PH-T	NPH-T
McNickleet al., 2019	69	69	35 (±1.5)	36.3 (±1.6)	56 (88.9)	53 (84.1)	13.1 (±0.8)	12.3 (±0.9)	3.2 (±0.1)	3 (±0.1)	-	-	110 (±4)	100 (±3)	126 (±4)	130 (±3)
Smith et al., 2018	127	77	31.3 (±0.7)	31.2 (±1.6)	111 (87.4)	68 (88.3)	9 (±0.5)	10.1 (±0.6)	2.8 (±0.2)	2.7 (±0.2)	-	-	100 (±2)	104 (±5)	114 (±2)	98 (±4)
Scerbo et al., 2017 *	252	29	33 (25.46) ^(1)^	34 (24.50) ^(1)^	212 (84.1)	27 (93.1)	9 (5.17) ^(1)^	20 (9.27) ^(1)^	3 (2.3) ^(1)^	3 (3.4) ^(1)^	15 (14.15) ^(1)^	14 (3.15) ^(1)^	100 (84.120) ^(1)^	122 (87.135) ^(1)^	119 (92.139) ^(1)^	100 (83.113) ^(1)^
Teixeira et al., 2018	181	845	34.4 (±1.1) ^(2)^	35.9 (±0.5) ^(2)^	157 (86.7)	708 (83.7)	13.2 (±0.8) ^(2)^	11.3 (±0.3) ^(2)^	36 (180) ^(3)^	77 (9.1) ^(3)^	28 (178) ^(4)^	91 (838) ^(4)^	105.9 (±2.1) ^(2)^	92.6 (±0.9) ^(2)^	125.3 (±7) ^(2)^	121.7 (±1.2) ^(2)^

PH-T: pre-hospital tourniquet; NPH-T: non pre-hospital tourniquet; ISS: Injury Severity Score; AIS: Abbreviated Injury Scale; GCS: Glasgow Coma Scale; HR: Heart Rate, SBP: systolic blood pressure; ^(1)^ expressed as median (IRQ range); ^(2)^ expressed as mean (SD); ^(3)^ expressed as N (extremity AIS ≥4); and ^(4)^ expressed as N (GCS < 8); * Scerbo et al. (2017) compared pre-hospital tourniquet vs. trauma centre tourniquet.

**Table 3 ijerph-18-12861-t003:** Outcome data for the comparisons of mortality.

Study	Pre-Hospital Tourniquet			No Pre-Hospital Tourniquet			*p*-Value	Time Point	Adjustment
	Sample	N Events	%	Sample	N Events	%			
McNickle et al. (2019)	69	0	0	69	2	2.9	NS	NR	Variable matching by patient demographics and injured artery, ISS, and mechanism of injury.
Smith et al. (2018)	127	9	7.1	77	10	13	0.21	NR	Variable matching by patient demographics and injury severity.
Teixeira et al. (2018)	181	7	3.9	845	44	5.2	0.45	NR	ISS, presence of associated severe head or torso injury, presence of major vascular injury, and traumatic amputation.
Study	Pre-Hospital Tourniquet			Trauma Centre Tourniquet			*p*-value	Time point	Adjustment
	Sample	N events	%	Sample	N events	%			
Scerbo et al. (2017)	252	13	5.2	29	4	13.8	0.07	NR	Data not adjusted

**Table 4 ijerph-18-12861-t004:** Transfusion of blood products.

(1) Packed Red Blood Cells Transfusion (pRBC)							
Studies	Pre-Hospital Tourniquet		No Pre-Hospital Tourniquet		*p*-Value	Time Point (Hours)	Adjustment
	Sample	Mean (SD)	Sample	Mean (SD)			
McNickle et al. (2019)	69	3.5 (0.5)	69	2.7 (0.8)	NS	within first 24 h	Variable matching by patient demographics and injured artery, and mechanism of injury.
Teixeira et al. (2018)	181	5.0 (8.6)	845	3.9 (14.5)	0.380	within first 24 h	Variable adjusted by age, sex, mechanism of injury, hypotension on admission, GCS, ISS, presence of associated severe head or torso injury, presence of major vascular injury, and traumatic amputation.
Smith et al. (2018)	127	2.0 (0.1) ^(1)^	77	9.3 (0.6) ^(1)^	<0.001	within first 24 h	Variable matching by patient demographics and injury severity.
Study	Pre-Hospital Tourniquet		Trauma Centre Tourniquet		*p*-value	Time point (hours)	Adjustment
	Sample	Median (IQR)	Sample	Median (IQR)			
Scerbo et al. (2017)	252	3 (1.6)	29	4 (2.9)	0.10	within first 24 h	
(2) Platelets transfusion							
Study	Pre-Hospital Tourniquet		No Pre-Hospital Tourniquet		*p*-value	Time point (hours)	Adjustment
	Sample	Mean (SD)	Sample	Mean (SD)			
Teixeira et al. (2018)	181	0.8 (2.2)	845	0.5 (2.4)	0.237	within first 24 h	Variable adjusted by age, sex, mechanism of injury, hypotension on admission, GCS, ISS, presence of associated severe head or torso injury, presence of major vascular injury, and traumatic amputation.
Study	Pre-Hospital Tourniquet		Trauma Centre Tourniquet		*p*-value	Time point (hours)	Adjustment
	Sample	Median (IQR)	Sample	Median (IQR)			
Scerbo et al. (2017)	252	1 (1.3)	29	2 (1.6)	0.11	within first 24 h	
(3) Plasma transfusion							
Studies	Pre-Hospital Tourniquet		No Pre-Hospital Tourniquet		*p*-value	Time point (hours)	Adjustment
	Sample	Mean (SD)	Sample	Mean (SD)			
Teixeira et al. (2018)	181	2.8 (6.8)	845	1.8 (4.7)	0.030	within first 24 h	Variable adjusted by age, sex, mechanism of injury, hypotension on admission, GCS, ISS, presence of associated severe head or torso injury, presence of major vascular injury, and traumatic amputation..
Smith et al. (2018)	127	1.4 (0.1) ^(1)^	77	6.2 (0.4) ^(1)^	<0.001	within first 24 h	Variable matching for patient demographics and injury severity
Study	Pre-Hospital Tourniquet		Trauma Centre Tourniquet		*p*-value	Time point (hours)	Adjustment
	Sample	Median (IQR)	Sample	Median (IQR)			
Scerbo et al. (2017)	252	3 (2.5)	29	5 (3, 10)	<0.01	within first 24 h	

^(1)^ expressed as mean (standard error); GCS: Glasgow Coma Scale; ISS: Injury Severity Score; NS: not statistically significant; IQR: interquartile range.

## Data Availability

All data generated or analysed during this study are included in this published article and its additional files, https://osf.io/n526s/ (accessed 29 November 2021).

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
