# Peer review of "Effectiveness of Pre-Hospital Tourniquet in Emergency Patients with Major Trauma and Uncontrolled Haemorrhage: A Systematic Review and Meta-Analysis"

_ijerph, 2021, doi:10.3390/ijerph182312861_

Round 1

Reviewer 1 Report

Thank you for the opportunity to review your paper about the Effectiveness of pre-hospital tourniquet in emergency patients with major trauma and uncontrolled haemorrhage: A systematic review and meta-analysis. Congratulations on your article. Below you can find some of my suggestions that can further improve the quality of the article.

In my opinion the article abstract is well structured and contains the necessary information. I only suggest improving the background with more information about de problematic. I also suggest that the authors improve the description of the methodology, referring to the main steps taken during the review. The conclusions must also be improved.

I also suggest that the keywords are according to the MESH terms.

The introduction is well written and properly contextualizes the review. The authors also present the review objectives at the end of the Introduction.

Regarding the methodology section, in my opinion the systematic literature review should only include primary studies and not literature reviews.

The search interval ended in March 2020. It has already elapsed more than a year after this date. I suggest the authors update the research to a more recent period.

In my opinion the article selection diagram should be explained and presented in the methodology section and not in the results section.

In my opinion, the characteristics of the studies should be summarized in just one table. Tables should be reconfigured, so they don't get too big.

The meta-analysis performed seems to me to be adequate.

The discussion is too dense (the authors present practically a discussion paragraph). I suggest dividing the text to be more objective and easier to read.

I also suggest that the authors focus more on the implications for practice of this review.

Author Response

Dear Reviever See the file

Reviewer 2 Report

Thank you for the opportunity to review this interesting manuscript. The authors are to be appreciated for their efforts on exploring the effectiveness of pre-hospital tourniquet use on major trauma patients and uncontrolled haemorrhage.

In my opinion, the authors provided an interesting report. The objectives were clearly stated. The necessities of the analysis were adequately explained. The study method was adequately described. The results clearly presented. The discussion pointed out the important findings. The conclusions appropriately based on the results and discussions.

I am appreciated for authors’ achievement. However, some concerns, as the authors mentioned, had raised from this work. Very few studies were included in the review. Only retrospective observational studies were included and analyzed. Selection bias and heterogeneous were also major concerns for these studies. It was difficult to provide solid evidence to answer the questions of this study. However, this study raised an important issue for the management of trauma patients. Though the authors had proposed some good suggestions, I am appreciated if the authors might provide further policy implications and more suggestions for future researches.

Author Response

Dear Revisor, please see the file
